# Residual Birch Wood Lignocellulose after 2-Furaldehyde Production as a Potential Feedstock for Obtaining Fiber

**DOI:** 10.3390/polym13111816

**Published:** 2021-05-31

**Authors:** Maris Puke, Daniela Godina, Mikelis Kirpluks, Janis Rizikovs, Prans Brazdausks

**Affiliations:** 1Latvian State Institute of Wood Chemistry, Dzerbenes 27, LV-1006 Riga, Latvia; daniela.godina@kki.lv (D.G.); mikelis.kirpluks@kki.lv (M.K.); janis.rizikovs@kki.lv (J.R.); prans.brazdausks@kki.lv (P.B.); 2Department of Chemistry, University of Latvia, Jelgavas 1, LV-1004 Riga, Latvia

**Keywords:** birch wood, pre-treatment, lignocellulose, 2-furaldehyde, monosaccharides

## Abstract

From birch wood, it is possible to obtain both acetic acid and 2-furaldehyde as valuable value-added products. The main objective of this study was to develop a new wasteless technology for obtaining 2-furaldehyde, acetic acid, and lignocellulose (LC) residue usable as feedstock in further processing such as thermomechanical (TMP), alkaline peroxide mechanical (APMP), and sulfate pulping processes. To achieve this objective several screening tests were performed, and a further experimental plan was developed using DesignExpert11. Process yields were analyzed both in terms of total yield and at individual time increments. In addition, the obtained LC residue was also characterized. A unique bench-scale reactor system was used to obtain an LC material without pentoses and with maximum preservation of cellulose fiber for further research. Studies on the deacetylation and dehydration of birch wood hemicelluloses of pentose monosaccharides to 2-furaldehyde and acetic acid using orthophosphoric acid as a catalyst were carried out. Results showed that, depending on the used pre-treatment conditions, the 2-furaldehyde yield was from 0.04% to 10.84% oven dry mass (o.d.m.), the acetic acid yield was from 0.51% to 6.50% o.d.m., and the LC residue yield was from 68.13% to 98.07% o.d.m. with minimal content of admixtures. Process optimization using DesignExpert11 revealed that the main pre-treatment process parameters that influenced the yield of 2-furaldehyde in the pre-treatment process were process temperature (53.3%) and process duration (29.8%).

## 1. Introduction

As oil plays a significant role in the global economy, the rapidly growing consumption of fossil energy resources and the overall decline in oil reserves led to the global energy crisis [1]. The growing demand for energy resources, the reduction of fossil energy stocks [2,3], and the need to control the quality of the environment put forward sustainable principles of development as one of the main directions of the future. Therefore, research is necessary for the development and industrialization of a new raw material base for the chemical and energy industries. Biorefinery as a concept refers to the usage of biomass acquired from renewable resources to produce energy and various chemicals. Lignocellulosic (LC) biomass is becoming a logical alternative to petroleum in light of looming oil shortages, increases in oil prices, and environmental sustainability considerations [2,4]. LC biomass refers to any materials which are rich in cellulose (40–50 wt%), hemicellulose (10–25 wt%), and lignin (25–40 wt%) [5,6]. For example, agriculture residues (sugarcane bagasse, corn stover, rice straw, etc.), forest products (e.g., wood, logging residues, shrubs), and herbaceous and woody energy crops are typical sources of LC biomass [5]. All these materials originally result from the biological photosynthesis from readily available atmospheric CO_2_, water, and sunlight [7]. Therefore, LC biomass is a sustainable and green feedstock for the generation of liquid fuels with net zero carbon emission that can eventually replace feedstock derived from petrochemical resources [8,9,10]. The transformation of biomass to useful chemicals is a major challenge for green chemistry [11,12,13]. Biofuel is obtained from renewable natural materials, which can be used as a substitute for petroleum sources. One of the most common biofuels is bio-ethanol, mainly produced from corn, wheat, or sugar beet. This can be defined as a first-generation biofuel, since classical food crops are used for the feedstock, the production of which requires high-quality agricultural land. The further development of first-generation biofuels raises the dilemma of food versus fuel in the utilization of available farmland. As such, more research is necessary on the development of processing pathways in which non-food feedstock is processed into biofuels, defined as second-generation biofuels. Birch chips are one promising resource suitable for obtaining various biofuels [14,15].

Among the major operations involved, the pre-treatment of LC biomass is the most important stage of the further biorefining concept. The mechanical structure of the LC biomass cell wall is changed during the pre-treatment process [16,17,18]. 

Currently, 2-furaldehyde is a valuable product for the synthesis of various kinds of chemicals and polymers. 2-Furaldehyde is a precursor for solvents, plastics, and food, pharmaceutical, and agricultural industries. It has been identified as one of the top 30 bio-based chemicals [19].

Consequently, it is very important to combine the production of 2-furaldehyde and cellulose fiber processes, but this has not yet been possible, even theoretically. In all known technologies for 2-furaldehyde production, a great proportion (40–50%) of the cellulose is destroyed during the pre-treatment process [20]. Therefore, it is important to elaborate theoretical grounds for a new lignocellulosic pre-treatment process, retaining cellulose in the LC residue for the further obtainment of thermomechanical (TMP), alkaline peroxide mechanical (APMP), and sulfate pulping processes, without cellulose losses.

It is possible to obtain 2-furaldehyde, acetic acid, and bioethanol from deciduous wood using a new invention as outlined in the patent “Method for obtaining 2-furaldehyde and ethanol” [21].

In Latvia’s climatic conditions, birch wood can be used as a raw material for the obtainment of 2-furaldehyde and acetic acid as well as TMP, APMP, and sulfate pulping. Birch stands occupy 30% of the Latvian territory’s forests (884 thousand ha), and are a good source of cellulose and hemicellulose raw materials [22]. The Laboratory of Biorefinery of the Latvian State Institute of Wood Chemistry has an original bench-scale hydrolysis reactor system for obtaining 2-furaldehyde, acetic acid, and LC residue, on which it is possible to study this process (Figure 1).

The scientific goal of this study was to carry out interdisciplinary research for the processing of birch wood in 2-furaldehyde and acetic acid, while simultaneously preserving the LC residue so that it can be used in pulp production. This would show that these two different technological processes can be combined into one biorefinery technology chain that has not been possible so far.

The technology will comprise a deciduous wood pre-treatment process, as a result of which valuable products (2-furaldehyde, acetic acid, and LC) will be obtained. 2-Furaldehyde and acetic acid are commercially realizable final products. In its turn, the remaining LC will be processed to obtain fiber.

The essential changes in the technological process would be such that two currently separate processes will be combined in one complex technology, as demonstrated in Figure 1. This will make it possible to reduce the consumption of raw materials and energy.

In the near future, due to decreasing oil stock, hardwood may become a real alternative to oil as a raw material for the production of chemicals, fuels, and materials. The main chemical product currently produced from pentosans and polyuronides is 2-furaldehyde. However, the joint production of 2-furaldehyde and fibers has not previously been possible because, in all known technologies, 40–50% of cellulose was destroyed in the process of obtaining 2-furaldehyde obtaining. This problem was successfully solved, and the destruction of cellulose in the new 2-furaldehyde obtainment process is no more than 3–5%. Therefore, it we considered it important, using the new possibilities [20], to investigate the effect of temperature and other factors and parameters on the changes in the hardwood LC composition in the pre-treatment process. In the present study, we analyzed the obtained experimental results on the effect of temperature on the birch wood LC composition in the pre-treatment process.

The pre-treatment process was carried out in an original bench scale hydrolysis reactor system. The volume of the main reactor was 13.7 L and the permissible steam pressure was 1.2 MPa. Birch wood chips (BWCs) were treated in a steam flow, at varying temperature. It was proved that by varying the temperature in the specified range, the amount of xylose in the LC residue decreased, in terms of the oven-dry mass.

The cellulose content in the LC residue under these conditions increased. Changes in the composition and amount of other monosaccharides in the LC residue depending on temperature were analyzed. The obtained results are important for optimizing the hardwood pre-treatment process.

## 2. Materials and Methods

In this section of the paper, we provide a detailed description of the process for obtaining the analyzed samples, the experimental design, the orthophosphoric acid-catalyzed pre-treatment method, and the high-performance liquid chromatography (HPLC) analysis. Initially, trial experiments (in total 13; E1–E13) were conducted to optimize the orthophosphoric acid-catalyzed hydrolysis process conditions (temperature, catalyst concentration, catalyst amount from the oven dry mass (o.d.m.), treatment time) for the birch wood chip C-6 degradation analysis. Based on the obtained data and observations from the trial experiments, a full factorial experimental plan of the work was created and developed. The new experimental work plan and the data obtained were processed in the computer program DesignExpert11.

### 2.1. Materials and Chemicals

Orthophosphoric acid (85%), sulfuric acid (95–97%), D-(+)-cellobiose (≥99%), D-(+)-glucose, (≥95%), D-(+)-xylose (≥99%), L-(+)-arabinose (≥99%), D-(+)-galactose (≥99%), D-(+)-mannose (≥99%), 2-furaldehyde (≥99%), acetic acid (≥99%), 5-hydroxymethylfurfural (≥99%), levulinic acid (≥98%), and formic acid (≥95%) were purchased from Merck and used without further purification.

### 2.2. Samples

BWCs were supplied by the A/S Latvijas Finieris company “Lignums”, focusing on the production of plywood and processed wood chips. The company supplies birch wood chips to pulp producers in Scandinavia. After they were obtained, wood chips were air dried and stored at 20 °C to prevent microbiological degradation prior to use. Birch wood chips were of particle size 45–47 mm.

### 2.3. Experimental Design

To be able to start our studies, based on the previous scientific experience and information available in the literature, the following limits of the pre-treatment process parameters variables were set, after the implementation of which it was possible to judge the direction in which to continue the experimental work (Table 1). In turn, the constant factor was the moisture of the raw material (w), 8%.

Using the computer program DesignExpert11, the initial full factorial experimental plan was developed, after the implementation of which it was possible to decide in which direction to continue the experimental studies and find the optimal process parameters for the pre-treatment process. Experimental work was performed on the bench equipment for 13 different experiments (E1–E13). Sixty-five condensate samples containing 2-furaldehyde, acetic acid, and other compounds were obtained and analyzed by HPLC (Shimadzu 20AD). Thirteen samples of LC residue were obtained, and their chemical composition was determined.

### 2.4. Catalyzed Pre-Treatment of Birch Wood Chips

Birch wood chips with particle size of 45–47 mm and moisture content W_rel_ = 8.71% were mixed in a catalyst solution in a blade-type mixer of special design. Orthophosphoric acid was used as a catalyst. After mixing the chips with a definite amount of the catalyst, the obtained material was treated with a continuous superheated steam flow in an original pilot plant. Three products were obtained—2-furaldehyde, acetic acid containing condensate, and LC leftover. The diameter of the main reactor was 110 mm, it height was 1450 mm and it had a volume of 13.7 L (Figure 2).

The reactor had two heat insulation systems with automatic equipment to ensure a constant temperature in the reaction zone during the whole process time and with different process parameters. The steam leaving the reactor, which contained mainly a water solution of 2-furaldehyde and acetic acid, was condensed, and samples were taken every 10 min. The steam-treated wood chips (LC) were discharged from the reactor. The chemical composition of the birch chips was determined by wet chemistry analytical standard methods as described in The Technical Association of the Pulp and Paper Industry (TAPPI) standards. All yields of the products and catalyst amounts were calculated from the o.d.m. For each sample, three parallel experiments were carried out, and the obtained results are shown as the average arithmetic result, with the relative standard deviation (RSD) for all experiments being less than 5%.

### 2.5. HPLC Analysis

The contents of monosaccharides, 2-furaldehyde, 5-hydroxymethylfurfural (5-HMF), and organic acids in the obtained hydrolysates were determined using a Shimadzu LC-20A HPLC (Shimadzu, Tokyo, Japan) with a refraction index detector. Cellobiose, glucose, xylose, arabinose, galactose, mannose, 2-furaldehyde, acetic acid, 5-HMF, levulinic acid, and formic acid (Sigma-Aldrich, Germany) with purity ≥ 99.0% were used as reference standards. For the cellobiose, glucose, 2-furaldehyde, acetic acid, 5-HMF, levulinic acid, and formic acid, we used a Shodex Sugar SH1821 column at 60 °C, with eluent 0.008 M H_2_SO_4_ at a flow rate of 0.6 mL·min^−1^. For the carbohydrate analysis we used a Shodex Sugar SP0810 column at 80 °C, with deionized water as the mobile phase under a flow rate of 0.6 mL·min^−1^. Samples were neutralized to pH 5–7 with NaHCO_3_ and filtered through a 0.45 μm membrane filter before injection. All samples were tested three times.

For each analyzed standard, the equations of the calibration curves are given in Table 2.

## 3. Results and Discussion

### 3.1. Analysis of the Raw Material

The chemical composition of the birch wood chips was determined as reported in Table 3.

The chemical composition of the used material is similar to information found in the literature [23]. Initial characterization of the birch wood chips indicates that this it is a promising feedstock suitable for further valorization due to the high amount of glucose and xylose. Xylose is a feedstock in the catalyzed pre-treatment process to obtain 2-furaldehyde and acetic acid, while the glucose-containing LC residue can be used in further processing into fiber materials. From the obtained results it is possible to calculate that the maximum theoretical amount of 2-furaldehyde obtainable from the BWCs is 16.45% from o.d.m.

### 3.2. Selection of the Initial Pre-Treatment Process Parameters for the Experimental Plan

Our previous studies have shown that the main parameters of the hemicellulose deacetylation and pentose monosaccharide dehydration process which influence the process dynamics (yield per unit of time—in this study, the amount every 10 min), yield (integral or total yield), and retained cellulose in the LC residue are the catalyst concentration and amount, the temperature, and the process duration [24]. Based on previous scientific experience and the information available in the literature [20,21,24], the following limits of the pre-treatment process parameters were set. Hydrolysis process temperature (T) from 130 to 180 °C, catalyst (H_3_PO_4_) concentration (c) from 10% to 95%, process duration (τ) from 10 to 90 min, steam flow rate in the reaction zone (v) from 100 to 240 g·min^−1^, and the catalyst amount (m) from 3% to 10%, calculated on the o.d.m. In turn, the constant factor was the moisture of the raw material (w).

The effects of the variable pre-treatment process parameters on 2-furaldehyde and acetic acid formation are listed in the Table 4. The yields of admixtures such as formic acid, levulinic acid, and 5-HMF were also determined, since higher pre-treatment process temperatures lead to the faster depolymerization of glucose and its irreversible degradation to by-products such as oxymethylfurfural, levulinic acid, and formic acid.

The yield of 2-furaldehyde was in the range of 0.04% to 10.84% in terms of the o.d.m.—that is, 0.21% to 65.87% from the theoretical 2-furaldehyde amount.

When considering acetic acid and 2-furaldehyde yields at individual time points (10 min increments) (Figure 3) it is noticeable that the highest number of desired products were obtained at the 20 min point, meaning that for both acetic acid and 2-furaldehyde most of the yield was obtained at the beginning of the process, in the first 50 min. After this point, obtained yields noticeably decreased. For 2-furaldehyde, the highest yield was obtained at 20 min. This was most noticeable for experiments E7, E1, and E3. Experiments E2, E5, E8, E9, E11, and E12 are not shown in the dynamic yield or total yield graphs since the total time for these experiments was only 10 min, meaning only 1 data point could be obtained for visualization. The highest yields of 2-furaldehyde and acetic acid were obtained in experiment E7 with the following experimental conditions: catalyst concentration, 10%; process temperature, 180 °C; catalyst amount, 10%; process duration, 90 min; and steam flow rate, 240 g·min^−1^. In Figure 3 and Figure 4A it can be seen that experiment E7 had the highest dynamic 2-furaldehyde and acetic acid yields. It shows that a high 2-furaldehyde content could be obtained with a relatively shorter treatment time, which could positively influence the quality of the obtained residual LC after pre-treatment.

As can be seen from the diagram generated with DesignExpert11 (Figure 5) the main pre-treatment process parameters that influenced the yield of 2-furaldehyde in the pre-treatment process were temperature (53.3%), process duration (29.8%), catalyst concentration (1.3%), catalyst amount (0.9%), and steam flow rate in the reaction zone (0.2%).

Based on the process results, the following process parameter Equation (1) was obtained:2-furaldehyde = −8.72186 + 0.013200(**c**) + 0.119701(m) + 0.053121(T) − 0.173083(τ) + 0.000037(v) + 0.001492(T·τ)(1)

For this prediction model, the standard deviation is 0.76 with R² of 0.9856.

The obtained equation can be used to make predictions about the response in furaldehyde yield for given levels of each factor. Here, the levels should be specified in the original units for each factor. This equation should not be used to determine the relative impact of each factor because the coefficients are scaled to accommodate the units of each factor and the intercept is not at the center of the design space.

As can be seen from Figure 6, the obtained 2-furaldehyde in the pre-treatment process experiments was comparable to values predicted with DesignExpert11, with a small margin of error.

For the acetic acid, the yield was in the range of 0.51% to 6.50% from o.d.m., which is 5.44% to 96.94% of the theoretical acetic acid amount. The pre-treatment process parameters for the birch wood deacetylation of the hemicelluloses are similar to pentose monosaccharide dehydration. The parameters of the pre-treatment process that had a greater impact on the deacetylation process were temperature and treatment time (Figure 7).

Based on the process results, the following Equation (2) was obtained:Acetic acid = −6.22382 + 0.003083(**c**) + 0.101082(m) + 0.045696(T) + 0.039672(τ) + 0.000052(v)(2)

For this prediction model, the standard deviation is 0.43 with an R^2^ of 0.9740.

Similar to the equation for 2-furaldehyde yield, here the levels should be specified in the original units for each factor.

As can be seen from Figure 8, the obtained acetic acid in the pre-treatment process experiments was comparable to the amounts predicted with DesignExpert11, with a small margin of error.

The obtained LC fraction yields (Figure 9) were in a range from 69% to 97% from o.d.m., with varying degree of purity as shown in Figure 10. The obtained yields of LC indicate that this pre-treatment can be successfully integrated in further processing pathways for obtaining polymeric LC fiber.

In terms of the relative ratios of the components of the obtained LC residue (lignin, glucan, and xylan) (Figure 10), more pronounced variations were noticeable depending on the pre-treatment conditions—process temperature and duration. When comparing obtained LC fractions with untreated biomass (Table 1), in all fractions there was a noticeable increase in lignin and glucan content due to a concentration effect connected with the separation of xylans and their subsequent upgrading to value-added products (i.e., 2-furaldehyde and acetic acid). When considering the obtained LC fractions as a feedstock for obtaining polymeric lignin and cellulose fiber, the LC residue with the highest content of lignin and glucan and minimal xylan admixtures was obtained in experiment E7, with the following experimental conditions: catalyst concentration, 10%; process temperature, 180 °C; catalyst amount, 10%; process duration, 90 min; and steam flow rate, 240 g·min^−1^. The data on 2-furaldehyde and acetic acid yield allow us to conclude that for successful pre-treatment, the most effective process utilizes the same conditions. This shows a potential for the obtained LC as a promising feedstock for further valorization and inclusion in biorefinery processing pathways.

## 4. Conclusions

In our study, depending on the pre-treatment conditions used, the 2-furaldehyde yield was from 0.04% to 10.84% o.d.m., the acetic acid yield was from 0.51% to 6.50% o.d.m., and the LC residue yield was from 68.13% to 98.07% o.d.m., with a minimal content of admixtures such as levulinic acid, formic acid, and 5-HMF.

Process optimization using DesignExpert11 revealed that the main pre-treatment process parameters that influenced the yield of 2-furaldehyde in the pre-treatment process were process temperature (53.3%) and process duration (29.8%).

The highest yield of 2-furaldehyde and acetic acid, and the LC residue with the highest content of lignin and glucan with minimal xylan admixtures, were obtained with following experimental conditions: catalyst concentration, 10%; process temperature, 180 °C; catalyst amount, 10%; process duration, 90 min; and steam flow rate, 240 g·min^−1^. The obtained results correlate with the values predicted with DesignExpert11. The best results were obtained at the maximum set limits of both process temperature and time, which were determined as critical points for process efficiency.

## Figures and Tables

**Figure 1 polymers-13-01816-f001:**
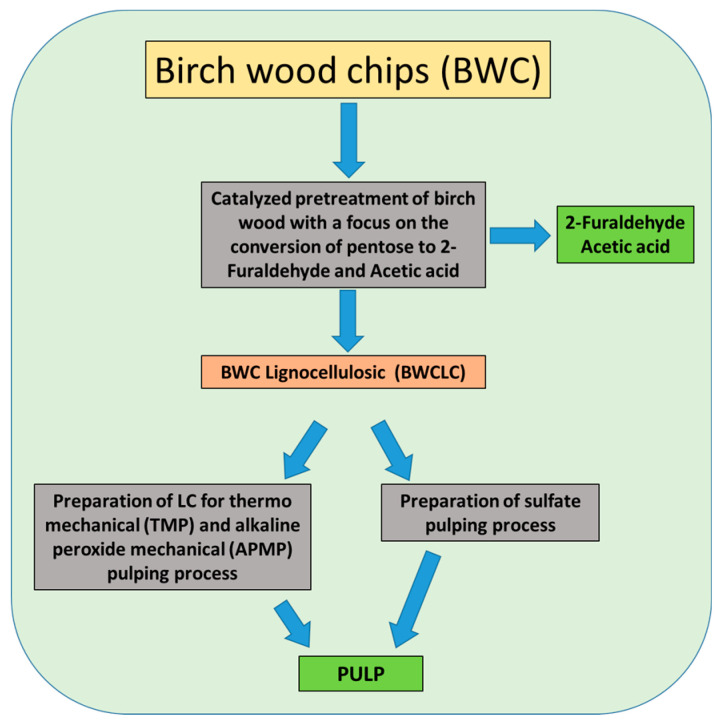
Basic flow diagram of the new process.

**Figure 2 polymers-13-01816-f002:**
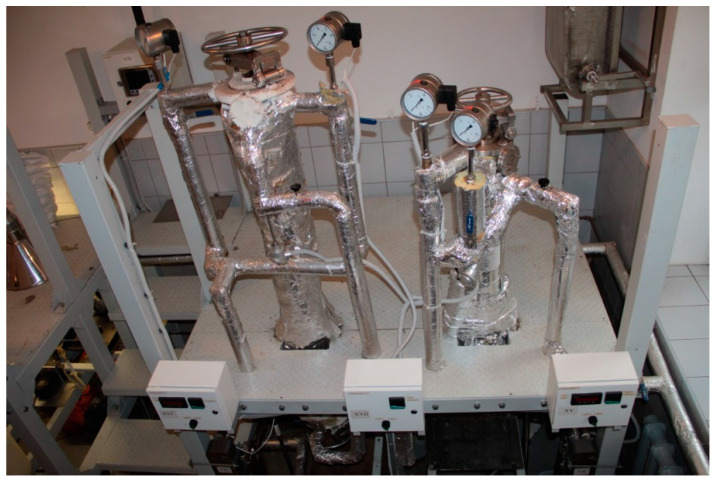
Bench-scale hydrolysis reactor system for obtaining 2-furaldehyde, acetic acid, and LC residue.

**Figure 3 polymers-13-01816-f003:**
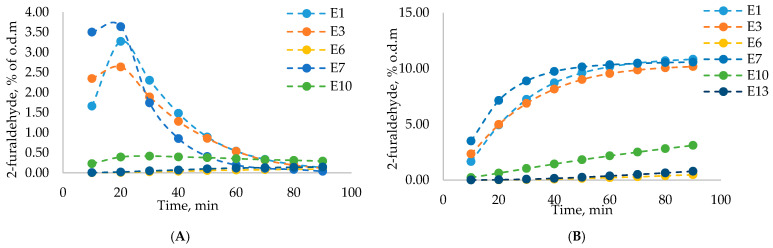
Dynamic 2-furaldehyde yield (% of o.d.m.) at individual time increments (**A**) and total 2-furaldehyde yield (% of o.d.m.) (**B**).

**Figure 4 polymers-13-01816-f004:**
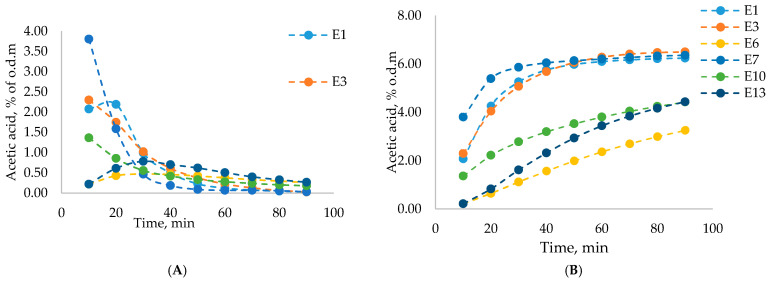
Dynamic acetic acid yield (% of o.d.m.) at individual time increments (**A**) and total acetic acid yield (% o.d.m.) (**B**).

**Figure 5 polymers-13-01816-f005:**
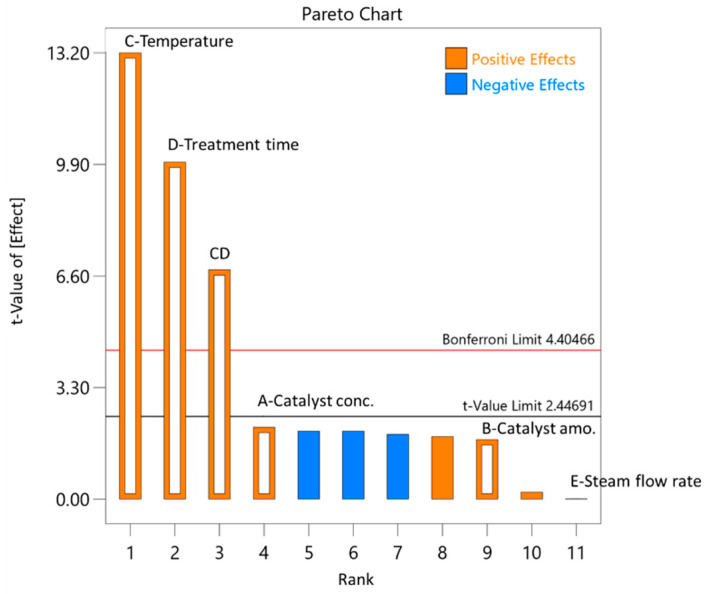
Influence of pre-treatment process parameters on 2-furaldehyde formation.

**Figure 6 polymers-13-01816-f006:**
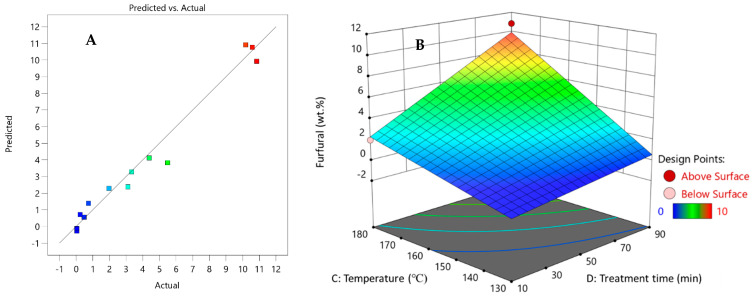
Parity plot of 2-furaldehyde (furfural) yield (**A**); influence of pre-treatment process parameters on 2-furaldehyde yield (**B**).

**Figure 7 polymers-13-01816-f007:**
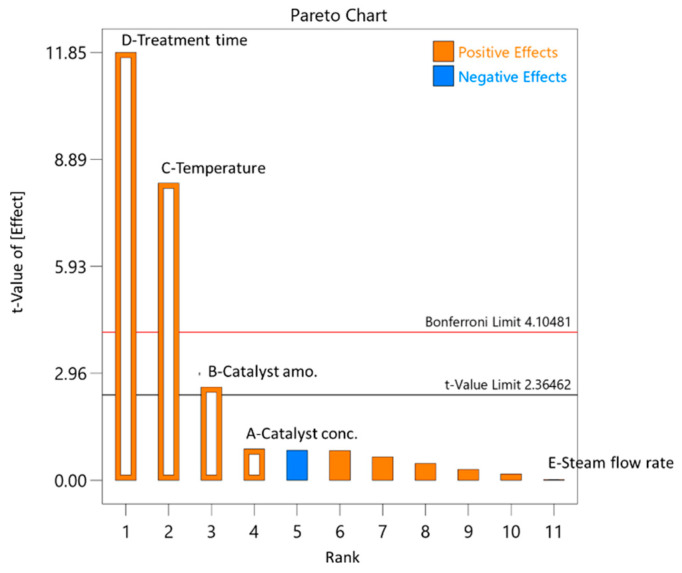
Influence of pre-treatment process parameters on acetic acid formation.

**Figure 8 polymers-13-01816-f008:**
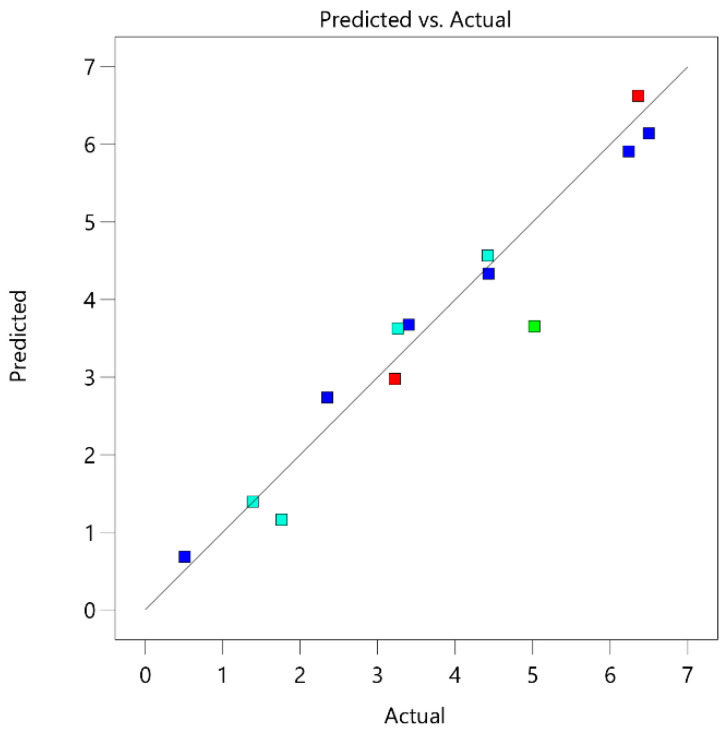
Parity plot of acetic acid yield.

**Figure 9 polymers-13-01816-f009:**
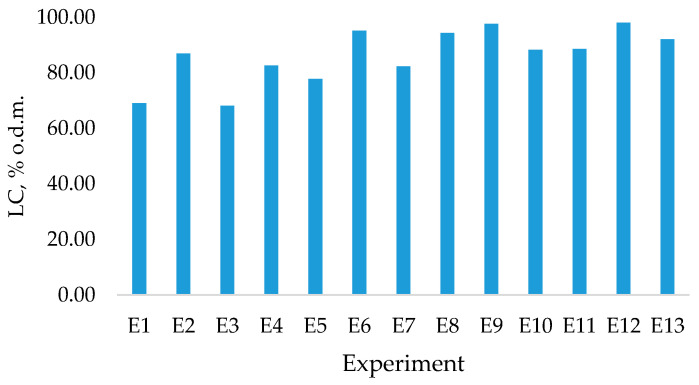
LC yield (% o.d.m.) from raw material.

**Figure 10 polymers-13-01816-f010:**
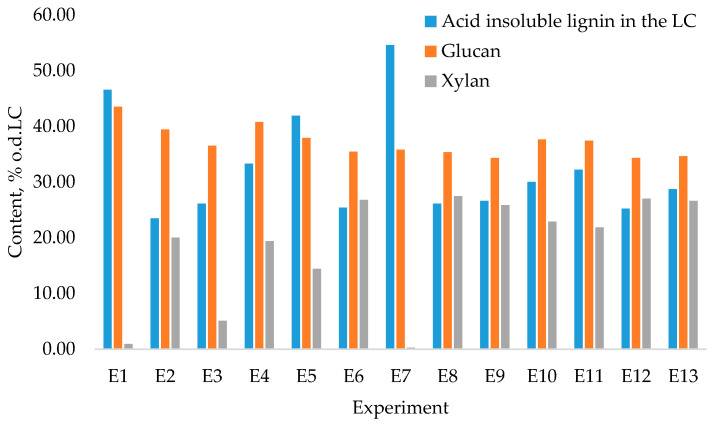
Content of major components in the obtained LC fractions (% o.d.LC).

**Table 1 polymers-13-01816-t001:** The pre-treatment process parameter variables.

Temperature	Catalyst (H_3_PO_4_) Concentration	Process Duration	Steam Flow Rate	Catalyst Amount
(T)	(c)	(τ)	(v)	(m)
130–180 °C	10–95%	10–90 min	100–240 g·min^−1^	3–10%

**Table 2 polymers-13-01816-t002:** Standard equations of the calibration curves.

Standard	Calibration Curve Equation	R^2^
Xylose	y = 1.34 × 10^11^x + 2188	0.99998
Arabinose	y = 1.33 × 10^11^x + 676	0.999990
Glucose	y = 1.42 × 10^11^x + 1491	0.99998
Galactose	y = 1.28 × 10^11^x	0.99994
Mannose	y = 1.45 × 10^11^x	0.99997
Cellobiose	y = 1.42 × 10^11^x	0.99994
2-Furaldehyde	y = 1.60 × 10^11^x − 927,950	0.999991
5-HMF	y = 1.67 × 10^11^x − 114,575	0.9999990
Acetic acid	y = 6.12 × 10^5^x − 631	0.999996
Levulinic acid	y = 9.56 × 10^5^x + 307	0.99998
Formic acid	y = 4.08 × 10^5^x + 803	0.999993

**Table 3 polymers-13-01816-t003:** Chemical composition of BWCs.

Compound	Amount (% from o.d.m.)
Extractives (ethanol-benzene)	4.24 ± 0.06
Extractives (hot water)	1.60 ± 0.40
Glucose	37.84 ± 0.05
Xylose	21.96 ± 0.06
Galactose	0.83 ± 0.05
Arabinose	0.66 ± 0.06
Mannose	1.60 ± 0.50
Acid-insoluble lignin	19.42 ± 0.04
Acid-soluble lignin	3.71 ± 0.06
Ash	0.60 ± 0.010
Other unidentified compounds	1.32 ± 0.05

**Table 4 polymers-13-01816-t004:** Effect of variable pre-treatment process parameters on 2-furaldehyde and acetic acid formation from birch wood chips.

Exp. No.	Variable Parameters, c/T/m/τ/ v	2-Furaldehyde Yield *	Acetic Acid Yield *	Formic Acid Yield *	Levulinic Acid Yield *	5-HMF Yield *	Lignocellulose Yield *
E1	LC/10/180/3/85/100	10.84 ± 0.06	6.24 ± 0.04	1.00 ± 0.04	0.041 ± 0.012	0.15 ± 0.02	69.09 ± 0.06
E2	LC/10/180/3/10/100	1.97 ± 0.02	2.35 ± 0.04	0.24 ± 0.02	0.031 ± 0.002	0.050 ± 0.012	86.98 ± 0.07
E3	LC/85/180/3/85/100	10.18 ± 0.04	6.50 ± 0.06	1.32 ± 0.07	0.17 ± 0.02	0.34 ± 0.05	68.13 ± 0.08
E4	LC/47.5/155/6.5/50/170	5.49 ± 0.03	5.02 ± 0.04	0.50 ± 0.04	0	0.031 ± 0.009	82.62 ± 0.05
E5	LC/85/180/10/10/100	4.39 ± 0.02	3.40 ± 0.02	0.39 ± 0.03	0	0.011 ± 0.002	77.85 ± 0.10
E6	LC/10/130/3/85/140	0.48 ± 0.03	3.26 ± 0.03	0.19 ± 0.02	0	0.030 ± 0.011	95.18 ± 0.12
E7	LC/10/180/10/85/240	10.59 ± 0.08	6.36 ± 0.05	1.52 ± 0.06	0.24 ± 0.02	0.41 ± 0.06	82.31 ± 0.06
E8	LC/85/130/10/10/140	0.24 ± 0.02	1.39 ± 0.04	0.10 ± 0.02	0	0.021 ± 0.008	94.34 ± 0.14
E9	LC/10/130/10/10/140	0.040 ± 0.010	1.76 ± 0.03	0.040 ± 0.010	0	0	97.68 ± 0.16
E10	LC/85/130/10/85/140	3.11 ± 0.04	4.42 ± 0.02	0.38 ± 0.04	0	0.010 ± 0.004	88.30 ± 0.10
E11	LC/85/180/3/10/240	3.33 ± 0.03	3.22 ± 0.07	0.38 ± 0.02	0.020 ± 0.011	0.041 ± 0.002	88.59 ± 0.06
E12	LC/85/130/3/10/100	0.039 ± 0.002	0.51 ± 0.04	0.040 ± 0.011	0	0	98.07 ± 0.07
E13	LC/10/130/10/85/100	0.79 ± 0.06	4.43 ± 0.08	0.26 ± 0.03	0	0	92.11 ± 0.16

* % from o.d.m.

## Data Availability

Not applicable.

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
