# Peer review of "Residual Birch Wood Lignocellulose after 2-Furaldehyde Production as a Potential Feedstock for Obtaining Fiber"

_polymers, 2021, doi:10.3390/polym13111816_

Round 1

Reviewer 1 Report

Reviewer comments for v1:

  This work conducted the hydrolysis of birch wood chips (BWCs) to transfer penta-saccharides to 2-furaldehyde, acetic acid and lignocellulose (LC) residue using ortho-phosphoric acid as a catalyst. The optimal results showed that the 2-furaldehyde yield was from 0.04 to 10.84% o.d.m., the acetic acid yield was from 0.51 to 6.50% o.d.m. and the lignocellulose residue yield was from 68.13 to 98.07 % o.d.m. However, there are some unclear places in the whole manuscript that need the authors to revise or explain. (1) From Sec. 3.1, the raw material is BWCs, thus, the caption of Table 2 “Chemical composition of raw material” should revise to “Chemical composition of BWCs” to make it more clearly. (2) From Table 2, I cannot understand why to sum up the total compounds are 115.03%? (3) Please make the nouns consistent in the whole text. For examples, “pretreatment” vs. “pre-treatment”; “lignocellulosic residue” vs. “lignocellulose residue”. (4) Please, the authors to abide by the following First writing problem to define o.d.m. and o.d.w. clearly in the main text. (5) In Fig. 6, the authors should note 2-furaldehyde is furfural.

From Table 2 (P.6)

Compound

Amount%, from o.d.m.

Extractives (ethanol-benzene)

4.24

Extractives (hot water)

1.60

Glucose

37.84

Xylose

21.96

Galactose

0.83

Arabinose

0.66

Mannose

1.60

Acid insoluble lignin

19.42

Acid soluble lignin

3.71

Ash

0.60

Acetyl groups

4.80

Other not identified compounds

1.32

2-furaldehyde (theoretical amount)

16.45

Total

115.03

Furthermore, this work has the central following two writing problems. First, in general, abbreviations should be defined the first time they appear in the abstract, and again the first time they appear in the main text. Once an abbreviation has been defined in the main text, it should be used exclusively throughout the rest of the manuscript (i.e., the full name does not need to write out again)., Please ensure that this and all other abbreviations are defined and used in this manner. Second, please have a blank space between data and unit except for data and %. The authors should correct, clear and concise the whole manuscript. I think the authors should revise the following problems word for word carefully and then re-submit this manuscript.

  1. 1 line 25, please have the full name of “o.d.m.” Please, the authors notice the above-mentioned First writing problem and review the full version like this.
  2. 1 line 25, “the acetic acid yield was from 0.51 to 6.50 %” should revise to “the acetic acid yield was from 0.51 to 6.50%”. Please, the authors notice the above-mentioned Second writing problem and review the full version like this.
  3. 2 line 64, “in the lignocellulose residue for the further obtainment of TMP” should revise to “in the lignocellulose (LC) residue for the further obtainment of TMP” due to LC is the first time that defined in the main text. Please, the authors notice the above-mentioned First writing problem and review the full version like this.
  4. 2 line 75, what is TRL4?
  5. 3 line 102, “Birch wood chips were treated in a steam flow” should revise to “Birch wood chips (BWCs) were treated in a steam flow” due to BWCs is the first time that defined in the main text. Please, the authors notice the above-mentioned First writing problem and review the full version like this.
  6. 3 in Fig. 1, correct “Birch wood chips (BWC)” to “Birch wood chips (BWCs)”.
  7. 4 line 116, please have a full name of HPLC.
  8. 4 line 116, “Initially trial experiments (in total 13) were done to optimize” should revise to “Initially trial experiments (in total 13; E1-E13) were done to optimize” due to make these experiments (E1-E13) more clearly in the following description.
  9. 4 line 131, “Birch wood chips was supplied by the…” should revise to “BWCs were supplied by the…” due to BWCs is previously defined in the main text and correct “was” to “were”.
  10. 4 line 140, correct “(Table 1.)” to “(Table 1)”. Please delete the redundant dot after Table 1.
  11. 4 line 149, “bench equipment for 13 different experiments” should revise to “bench equipment for 13 different experiments (E1-E13)”.
  12. 5 line 167, please have a full name of TAPPI.
  13. 5 line 176, please have a full name of HMF.
  14. 5 lines 177-178, “…determined using a high performance liquid chromatograph (HPLC) SHIMADZU LC-20A” should revise to “…determined using a HPLC SHIMADZU LC-20A” due to HPLC has been defined previously in the main text.
  15. 11 line 308, correct “bean” to “be”.
  16. 11 line 309, correct “predicted” to “predict”.
  17. 11 line 316, correct “figure 10” to “Figure 10”.
  18. 12 line 342, correct “DesignExpert 11” to “DesignExpert11” due to make this noun DesignExpert11 consistent in the whole text.
  19. The language should be polished, entirely by a native English speaker.

I hope my comments will be useful to improve the quality of your research and manuscript.

Author Response

Dear Editor,

We have revised the manuscript “Residual birch wood lignocellulose after 2-furaldehyde pro-duction as a potential feedstock for fibre obtaining” to make it suitable for publication in Polymers. First reviewer pointed out that our publication has the central following two writing problems. First, in general, abbreviations had to be defined the first time they appear in the abstract, and again the first time they appear in the main text. Second, we had to correct a blank space between data and unit except for data and %. Second reviewer suggested to expand the introduction part by including applications of wood for other fuel purposes. Also, the method of analysis by liquid chromatography lacked the equations of standard substances, and in one table at results the standard deviations were not given. We took notice of the reviewer’s comments and made all the necessary corrections.

We have attached a point-by-point response to the reviewer’s comments and hope that you will soon let us know about the acceptance of the manuscript.

Since the text was edited and introduction was supplemented, line numbers has changed. In bold are given the new corresponding line numbers.

Dear Reviewer (1),

Your comments about our manuscript “Residual birch wood lignocellulose after 2-furaldehyde pro-duction as a potential feedstock for fibre obtaining” were very useful and motivating. Thank you for recognising the work’s importance in the field of biomass processing. Here is the response to your comments:

  1. From Sec. 3.1, the raw material is BWCs, thus, the caption of Table 2 “Chemical composition of raw material” should revise to “Chemical composition of BWCs” to make it more clearly.

Suggestion considered and corrected in manuscript. Line 206.

  1. From Table 2, I cannot understand why to sum up the total compounds are 115.03%?

I agree with this comment. This situation happened since in the table results of the calculated maximal obtainable 2-furaldehyde yield was included, as well as acetyl group content. These two results can not be summed with other results. So, these results moved from the table to main text. Table should now present more clearer picture. Line 200.

  1. Please make the nouns consistent in the whole text. For examples, “pretreatment” vs. “pre-treatment”; “lignocellulosic residue” vs. “lignocellulose residue”.

In the manuscript the nouns “pre-treatment” and “lignocellulose residue” were done consistent. Lines 15, 24, 25, 27, 28, 30, 61, 63, 70, 71, 72, 83, 87, 90, 106, 108, 109, 113, 117, 118, 120, 124, 147, 182, 161, 162, 183, 211, 212, 215, 222, 228, 230, 234, 257, 260, 262, 263, 276, 285, 288, 290, 298, 315, 324, 329, 330, 340, 345, 347, 349, 350.

  1. Please, the authors to abide by the following First writing problem to define o.d.m. and o.d.w. clearly in the main text.

In the text we chose one of these two abbreviations – o.d.m. and it was defined in text as oven dry mass. Lines 25, 26, 178, 206, 214, 226, 234, 237, 238, 239, 240, 241, 287, 320, 346, 347.

  1. In Fig. 6, the authors should note 2-furaldehyde is furfural.

Corrected in the manuscript. Line 285.

  1. Line 25, please have the full name of “o.d.m.” Please, the authors notice the above-mentioned First writing problem and review the full version like this.

Corrected in the manuscript as oven dry mass. Line 25.

  1. Line 25, “the acetic acid yield was from 0.51 to 6.50 %” should revise to “the acetic acid yield was from 0.51 to 6.50%”. Please, the authors notice the above-mentioned Second writing problem and review the full version like this.

In the whole text we revised the Second writing problem. Line 25.

  1. Line 64, “in the lignocellulose residue for the further obtainment of TMP” should revise to “in the lignocellulose (LC) residue for the further obtainment of TMP” due to LC is the first time that defined in the main text. Please, the authors notice the above-mentioned First writing problem and review the full version like this.

Corrected in the manuscript. We also defined the abbreviation TMP as thermomechanical process. Line 72.

  1. Line 75, what is TRL4?

Corrected in the manuscript – it was a minor error; it was not necessary in the text. Line 84.

  1. Line 102, “Birch wood chips were treated in a steam flow” should revise to “Birch wood chips (BWCs) were treated in a steam flow” due to BWCs is the first time that defined in the main text. Please, the authors notice the above-mentioned First writing problem and review the full version like this.

We corrected suggested First writing problem and all others that appeared in the text. Line 111.

  1. In Fig. 1, correct “Birch wood chips (BWC)” to “Birch wood chips (BWCs)”.

Corrected in the manuscript. Line 115.

  1. Line 116, please have a full name of HPLC.

The full name of HPLC was given – high performance liquid chromatography. Line 125.

  1. Line 116, “Initially trial experiments (in total 13) were done to optimize” should revise to “Initially trial experiments (in total 13; E1-E13) were done to optimize” due to make these experiments (E1-E13) more clearly in the following description.

To make clearer vision of the experimental design we revised the sentence. Line 126.

  1. Line 131, “Birch wood chips was supplied by the…” should revise to “BWCs were supplied by the…” due to BWCs is previously defined in the main text and correct “was” to “were”.

Corrected in the manuscript. Line 140.

  1. Line 140, correct “(Table 1.)” to “(Table 1)”. Please delete the redundant dot after Table 1.

Corrected in the manuscript. Line 149.

  1. Line 149, “bench equipment for 13 different experiments” should revise to “bench equipment for 13 different experiments (E1-E13)”.

We revised this sentence. Line 158.

  1. Line 167, please have a full name of TAPPI.

The full name of TAPPI was given in the manuscript - The Technical Association of the Pulp and Paper Industry. Line 176.

  1. Line 176, please have a full name of HMF.

The full name of HMF was given in the manuscript – 5-hydroxymethylfurfural. Line 186.

  1. Lines 177-178, “…determined using a high performance liquid chromatograph (HPLC) SHIMADZU LC-20A” should revise to “…determined using a HPLC SHIMADZU LC-20A” due to HPLC has been defined previously in the main text.

Corrected in the manuscript. Line 187.

  1. Line 308, correct “bean” to “be”.

Corrected in the manuscript. Line 315.

  1. Line 309, correct “predicted” to “predict”.

Corrected in the manuscript. Line 316.

  1. Line 316, correct “figure 10” to “Figure 10”.

Corrected in the manuscript. Line 323.

  1. Line 342, correct “DesignExpert 11” to “DesignExpert11” due to make this noun DesignExpert11 consistent in the whole text.

In the whole text the “DEsignExpert11” noun was revised consistent. Lines 27, 349, 356.

  1. The language should be polished, entirely by a native English speaker.

The manuscript was given for proof reading.

Reviewer 2 Report

In order to improve the reception of the article and issues related to the use of lignocellulosic biomass, I would suggest expanding the literature to include applications of wood for other fuel purposes (for example, those covered in the articles: 

1. Roman, K.; Barwicki, J.; Hryniewicz, M.; Szadkowska, D.; Szadkowski, J. Production of Electricity and Heat from Biomass Wastes Using a Converted Aircraft Turbine AI-20. Processes 2021, 9, 364. https://doi.org/10.3390/pr9020364

2. Demirbas, A.; Competitive liqid biofuels from biomass. Applied Energy 2011, Vol. 88 issue 1, pp.17-28, DOI: https://doi.org/10.1016/j.apenergy.2010.07.016

3. Gomez, L.D.; Steele-King C.G.; McQueen-Mason S.J.; Sustainable liquid biofuels from biomass: the writing's on the walls. New Phytologist 2008, Vol. 178, Issue 3, pp. 473-485, DOI: https://doi.org/10.1111/j.1469-8137.2008.02422.x

4. Stocker, M.; Biofuels and biomass- to- liquid fuels in biorefinery: Catalytic conversion of lignocellulosic biomass using porous materials, Angewandte Chemie International Edition 2008, Vol. 47, Issues 48, pp. 9200-9211, DOI: https://doi.org/10.1002/anie.200801476

The method of analysis by liquid chromatography HPLC,(line 175 to 187) lacks the equations of the calibration curves of the standards used to convert the concentrations of the substances in the resulting solution. It would be necessary to supplement the methodology with this information. 

The standard deviations of the results obtained are missing from Table 3. 

It would be useful to supplement the results with a selected test of significance of the differences in the results obtained.  

Author Response

Dear Reviewer (2),

Your comments about our manuscript “Residual birch wood lignocellulose after 2-furaldehyde pro-duction as a potential feedstock for fibre obtaining” were clear and useful. Thank you for recognising the work’s importance in the field of biomass processing. Here is the response to your comments:

  1. In order to improve the reception of the article and issues related to the use of lignocellulosic biomass, I would suggest expanding the literature to include applications of wood for other fuel purposes (for example, those covered in the articles:
  2. Roman, K.; Barwicki, J.; Hryniewicz, M.; Szadkowska, D.; Szadkowski, J. Production of Electricity and Heat from Biomass Wastes Using a Converted Aircraft Turbine AI-20. Processes 2021, 9, 364. https://doi.org/10.3390/pr9020364
  3. Demirbas, A.; Competitive liqid biofuels from biomass. Applied Energy 2011, Vol. 88 issue 1, pp.17-28, DOI: https://doi.org/10.1016/j.apenergy.2010.07.016
  4. Gomez, L.D.; Steele-King C.G.; McQueen-Mason S.J.; Sustainable liquid biofuels from biomass: the writing's on the walls. New Phytologist 2008, Vol. 178, Issue 3, pp. 473-485, DOI: https://doi.org/10.1111/j.1469-8137.2008.02422.x
  5. Stocker, M.; Biofuels and biomass- to- liquid fuels in biorefinery: Catalytic conversion of lignocellulosic biomass using porous materials, Angewandte Chemie International Edition 2008, Vol. 47, Issues 48, pp. 9200-9211, DOI: https://doi.org/10.1002/anie.200801476

Introduction has been revised and additional information from suggested publications are given. Lines 52 – 60.

  1. The method of analysis by liquid chromatography HPLC, (line 175 to 187) lacks the equations of the calibration curves of the standards used to convert the concentrations of the substances in the resulting solution. It would be necessary to supplement the methodology with this information.

For all the analysed standard substance the equation of the calibration curves are given in a table, also we added the correlation coefficients (R2). Lines 197 – 201.

  1. The standard deviations of the results obtained are missing from Table 3.

The standard deviations of the obtained results from Table 3. are now given. Line 206.

  1. It would be useful to supplement the results with a selected test of significance of the differences in the results obtained.

Results have been supplemented with fit statistics of predicted model, Std dev and R2. Lines 271; 296.

Round 2

Reviewer 1 Report

Reviewer comments for v2:

  I think the authors have revised something in the manuscript. However, due to the authors have defined “lignocellulose” as “LC” on P.2 line 72. Therefore, according to previously First writing problems of v1 comments, the authors should revise this abbreviation (LC) in abstract and main text. After the careful revision performed by the authors, I think it can be accepted as it is.

  1. 1 line 15, “acetic acid and lignocellulose residue usable as feedstock…” should revise to “acetic acid and lignocellulose (LC) residue usable as feedstock…” due to LC is the first time that defined in the abstract.
  2. 1 line 19, “obtained lignocellulose residue was also” should revise to “obtained LC residue was also” due to LC has been defined previously in the abstract.
  3. 3 line 83, “acetic acid and lignocellulose residue on which it is” should revise to “acetic acid and LC residue on which it is” due to LC has been defined previously in the main text. The following text has occurred on the same problems: P.3 lines 87, 91-92, 105, 107, 113; P.4 lines 117-118; P.5 lines 161, 174; P.7 line 212; P.9 line 257; P.12 line 327; P.13 lines 331, 349.
  4. 5 lines 159-160, “…were obtained and analyzed by high- performance liquid chromatograph (HPLC - Shimadzu 20AD)” should revise to “…were obtained and analyzed by HPLC (Shimadzu 20AD)” due to HPLC has been defined previously in the main text.

Author Response

Dear Reviewer (1),

Your comments from the revision round 2 (v2) about our manuscript “Residual birch wood lignocellulose after 2-furaldehyde pro-duction as a potential feedstock for fibre obtaining” were considered. Here is the response to your comments:

  1. 1 line 15, “acetic acid and lignocellulose residue usable as feedstock…” should revise to “acetic acid and lignocellulose (LC) residue usable as feedstock…” due to LC is the first time that defined in the abstract.

Corrected in the text. Line 15

  1. 1 line 19, “obtained lignocellulose residue was also” should revise to “obtained LC residue was also” due to LC has been defined previously in the abstract.

Corrected in the text. Line 19

  1. 3 line 83, “acetic acid and lignocellulose residue on which it is” should revise to “acetic acid and LC residue on which it is” due to LC has been defined previously in the main text. The following text has occurred on the same problems: P.3 lines 87, 91-92, 105, 107, 113; P.4 lines 117-118; P.5 lines 161, 174; P.7 line 212; P.9 line 257; P.12 line 327; P.13 lines 331, 349.

As lignocellulose was defined in the main text for the first time (line 39), in the following text we corrected to LC. Lines 41; 45; 47; 60; 61; 70; 81; 85; 89; 90; 103; 105; 111; 115; 116; 159; 167; 173; 182; 211; 218; 256; 321; 323; 324; 326; 330; 332; 336; 337; 342; 348; 353

  1. 5 lines 159-160, “…were obtained and analyzed by high- performance liquid chromatograph (HPLC - Shimadzu 20AD)” should revise to “…were obtained and analyzed by HPLC (Shimadzu 20AD)” due to HPLC has been defined previously in the main text.

As high-performance liquid chromatography was defined in the main text (line 123), in the following text we corrected to HPLC. Lines 158; 186